



# Long-term changes in the ocean tide at Port Louis, Falkland Islands

Philip L. Woodworth[1]

[1]National Oceanography Centre, Joseph Proudman Building, 6 Brownlow Street, Liverpool, L3 5DA, United Kingdom

*Correspondence to*: Philip L. Woodworth (plw@noc.ac.uk)

**Abstract.** The historic tide gauge measurements at Port Louis in the Falkland Islands made by James Clark Ross in 1842 have been used to see whether there have been long-term changes in the ocean tide at that location. The conclusion is that there is no evidence for any significant change, which contrasts with tide gauge findings from other parts of the world over similar timescales. As by-products, the study has also been instructive in providing an example of how to obtain accurate tidal information from tabulations of high and low waters and from short tide gauge records.

## 1. Introduction

There is great interest in obtaining a better understanding of the changes in the ocean tide which have been observed at many locations around the world during the past century, especially their relationship to changes in ocean climate and sea level (Haigh et al., 2020). Many of these observations have come from tide gauges located along northern hemisphere coastlines, which have often been subject to engineering changes through the years. Therefore, there is particular interest in discovering whether there have also been long-term changes in the tide at pristine, remote locations, such as at islands in the southern hemisphere. Indeed, one might say that it was through such a study at a remote location (that of changes in the tide at St. Helena since 1761 by Cartwright, 1971, 1972) that this area of research began.

The present paper investigates whether there have been tidal changes since 1842 at Port Louis, located at the head of an inlet called Berkeley Sound on the east coast of East Falkland in the Falkland Islands in the SW Atlantic Ocean (Figure 1). Findings from the historical tidal observations of James Clark Ross have been compared to those obtained from a number of temporary tide gauge installations in the last few decades at Port Louis and nearby, with a focus on the predominant lunar semidiurnal tidal component (M2). As by-products, the study has provided lessons on how to obtain reliable tidal information from the processing of short tide gauge records and from records of high and low waters.

Section 2 describes the available tidal data sets, which are analysed in Sect. 3 to provide estimates of the historical and modern amplitudes and phase lags of the predominant M2 tidal component. The conclusions drawn from these separate analyses are given in Sect. 4.



## 2. Tidal Data Sets at Port Louis

### 2.1 The 1842 Data of James Clark Ross

James Clark Ross and the crews of his two ships (*Erebus* and *Terror*) had an extended stay in 1842 at Port Louis during their famous voyage of discovery and research in the Southern Ocean (Ross, 1847a,b). Ross installed a magnetic observatory at the site and undertook a set of astronomical, oceanographic and meteorological measurements. Tidal measurements were made between 10 May and 15 December, probably using a tide board (tide staff) or a form of float and stilling well gauge, with sea levels measured with respect to a nearby

benchmark. The measurements were made every half an hour, and more frequently around the times of every high and low water, although with major gaps between September and December when Ross took his ships to Tierra del Fuego to make simultaneous measurements of the magnetic field in South America and the Falklands. As a result, only the measurements up to the end of August are useful.

An understanding of just how much 'more frequently' the measurements were made around high and

low water will be important below. This is simply how Ross described them in Ross (1847b). However, in an undated letter to Sir George Biddell Airy headed "Copy of Note on Tides at Port Louis", held in box 499, number 177 in the archives of the Royal Greenwich Observatory at Cambridge University Library, Ross referred to them more precisely as 'every quarter of an hour'.

Ross's original journal, containing the entire set of half-hourly measurements at Port Louis and the

associated quarter-hourly values, is believed to have been destroyed accidentally by the UK Hydrographic Office (UKHO) during the 1990s. Fortunately, the journal was inspected in some detail by William Whewell and coworkers at Trinity College, Cambridge, during the nineteenth century, and a record of the heights and times of each high and low (HL) water was kept in Whewell's research notes that are preserved in the Wren Library. Those tabulations of heights and times were converted into computer files by the National

Oceanography Centre (NOC) and formed the basis of the study of changes in mean sea level at Port Louis since 1842 by Woodworth et al. (2010).

The fact that access to the original half-hourly measurements is no longer possible means that we cannot undertake a modern harmonic tidal analysis of the data. Fortunately, the Port Louis 1842 half-hourly record was analysed in the 19[th] century by Edward Roberts and published by Baird and Darwin (1885). These

scientists were very much experts in tidal analysis and their findings can be relied upon. In addition, the HL values of Whewell provide a separate source of information on the tide that we do nowadays have access to. It will be seen that, when one considers how Whewell (or more likely one of his assistants) came to make the HL tabulations, that findings from them are consistent with those published by Baird and Darwin many years ago.

The amplitude of M2 estimated from the HL waters has to be adjusted for nodal and seasonal

variations, given that the record spans only a short period in May-August 1842. The nodal adjustment assumes that the observed M2 amplitude varies over 18.6 years as in the equilibrium tide, which should be a reasonable assumption at an ocean island. The seasonal adjustment makes use of the annual amplitudes of variation in M2 estimated from each of 18 years of data from the permanent tide gauge at Port Stanley located 25 km southeast



of Port Louis. The tide at Port Stanley is similar in magnitude to that at Port Louis, and it was considered that
the two locations were near enough that their seasonal variations in M2 would be similar also.

**2.2 Recent Tide Gauge Deployments**

The six sets of recent tide gauge measurements, of which four consist of two pairs of simultaneous recordings at
slightly different locations, are listed in Table 1, and the locations of the tide gauges at the head of Berkeley
Sound are shown in Figure 1(c). Record 1 was obtained from a float gauge operated by the UKHO with data
recovered in 1982 by HMS Endurance following the Falklands war. All the other records were obtained from
sensors operated by NOC. The gauges included ones which delivered sea levels in centimetres (the float and
differential sensors) and ones which delivered absolute pressure values in mbar. Fortunately, conversion of one
parameter to the other, involving the product of water density times acceleration due to gravity, results in a
scaling factor of almost exactly 1.0. Therefore, cm and mbar can be considered to be the same unit at this
location. In principle, the M2 amplitudes and phase lags obtained from the absolute sensors should be adjusted
for the M2 tide in local air pressure, if they are to be taken as the true M2 ocean tide. However, the M2 air tide
is considerably smaller than any uncertainty in the tide gauge measurements (amplitude of the order of 1 Pa or
0.01 cm, see Figure 16.6 of Schindelegger et al., 2023). Further details of each tide gauge deployment can be
found in Woodworth et al. (2010).

**3. Comparison of Historical and Modern M2**

**3.1 Harmonic analysis of the 1842 data**

As mentioned above, the researchers of the 19$^{th}$ century had access to the original half-hourly data set of James
Clark Ross. This was analysed on behalf of the British Association by Edward Roberts in 1878 and his findings
were reported by Baird and Darwin (1885, pages 139 and 198) with only a simple change in the units used for
amplitude. This analysis will have included nodal adjustments as expected from the equilibrium tide (see page
136). The 4 major constituents in their listing of amplitudes (H) and phase lags (κ) were as follows:

| Constituent | H(ft) | (cm) | κ (°) |
|---|---|---|---|
| M2 | 1.544 | 47.1 | 157 |
| S2 | 0.492 | 15.0 | 195 |
| K1 | 0.358 | 10.9 | 37 |
| O1 | 0.451 | 13.7 | 4 |

with a ratio of N2 and M2 amplitudes of 0.217. The phase lags employed by Baird and Darwin (κ) refer to local
mean time at a particular longitude (58° W). Special Publication No. 26 (1963) of the International
Hydrographic Organization contained the same information as Baird and Darwin, giving the latter as its source
(with one misprint).

Between 1948 and 1984, the Admiralty Tide Tables (ATT) contained Port Louis amplitudes (H) and
phase lags (g) derived from Ross's data as follows:



| Constituent | H(cm) | g (°) | G(°) |
|---|---|---|---|
| M2 | 47 | 157 | 273 |
| S2 | 15 | 192 | 312 |
| K1 | 11 | 35 | 95 |
| O1 | 14 | 7 | 63 |


where this type of phase lag (g) refers to a particular timezone, being GMT+4 in this case. (This notation
follows the sign convention used in the ATT tables, meaning 4 hours behind Greenwich Mean Time.) These
values are almost the same as those of Baird and Darwin, but it is possible that the Hydrographic Office
undertook their own separate analysis of Ross's data, obtaining similar findings. If they simply copied Baird and
Darwin, it seems that 58° W must have been considered to be close enough to 60° W for the reported phase lags
to be taken as being in GMT+4 (hence κ and g being the same). These g values would then correspond to
Greenwich phase lags (G) as shown. The UKHO subsequently stopped making use of Ross's 1842 data, and, for
ATT purposes, employed harmonic constants obtained from measurements at Green Patch (record 1 in Table 1).

### 3.2 Use of 1842 HL values

#### 3.2.1 The Use of Mean Tidal Range

Mean Tidal Range (MTR) is defined by Mean High Water (MHW) minus Mean Low Water (MLW) and is
approximately twice the amplitude of M2. Averages of the 1842 HL values were 185.3 cm for MHW (from 218
individual high waters) and 80.3 cm for MLW (from 219 individual low waters), both being measured to the
same datum (which need not concern us here). Therefore, MTR is 105.0 cm implying a first estimate of M2
amplitude of 52.5 cm.

However, there are several contributions from other components of the tide which result in this
obtained MTR being larger than twice M2 amplitude. These contributions stem primarily from the other
semidiurnal tidal components (i.e. S2, N2, K2 etc.) and, to a lesser extent, from the diurnal components and
from harmonics of M2 (i.e. M4, M6 etc.). They can be computed readily from the results of harmonic analysis
of modern data as described by Doodson and Warburg (1941) and Woodworth et al. (1991, Appendix A). Of
course, this procedure assumes that these aspects of the tide have not changed since 1842. In this case, the
results of the 1984 tidal analysis show that the contributions amount to 7.6 cm, resulting in an adjusted MTR of
97.4 cm, or adjusted M2 amplitude of 48.7 cm.

Two further adjustments are necessary. One is due to the nodal variation of M2, the amplitude of which
in the equilibrium tide varies by 3.7% about its average value over a period of 18.6 years. An assumption that
the nodal variation at Port Louis is similar to that in the equilibrium tide is supported by inspection of the 18
years of M2 amplitude from Port Stanley, which shows no evidence for a residual nodal signal once amplitudes
are adjusted by the equilibrium amount (Figure 2a). The Ross measurements in mid-1842 occurred
approximately 5.75 years before nodal maximum for M2, and so nearer to nodal minimum than nodal
maximum. Therefore, the estimate of the average M2 amplitude has to be increased by 3.7 cos(2 π 5.75/18.6) or
1.34%, resulting in a further adjusted amplitude of 49.4 cm.

The second adjustment is due to the fact that M2 amplitude varies through the year. The same 18 years
of data from Port Stanley show that the amplitude of the annual variation in M2 there is typically 0.47 cm with
M2 amplitude peaking around the end of August or beginning of September (Figure 2b). An assumption that M2





at Port Louis varies seasonally in a similar way to that at Port Stanley then results in its average amplitude being further revised, downwards by 0.3 cm in this case, resulting in a final estimation of historical M2 amplitude of
approximately 49 cm.

### 3.2.2 Comparison to Tidal Predictions

In this second analysis of the HL information, a calculation was made of MHW-MLW (or MTR) from the 1842 HL values minus those from predictions based on the harmonic constants computed from tidal analysis of the 1984 data, assuming that all components of the tide were the same in 1842 as in 1984, except for the amplitude
of M2. In a first run, the M2 amplitude was assumed to be the same, but in subsequent runs it was allowed to vary in increments of 0.5 mm from that in 1984. No concerns about nodal variability are necessary in this case, as such variations are taken care of automatically in the computation of the predictions. Table 3(a) shows the findings, which suggest that the best fit to the 1842 value of MTR requires an M2 amplitude 2.25 cm larger than that used in the predictions based on 1984 data i.e. 2.25 + 48.04 = 50.29 cm.

160             A similar procedure was undertaken for M2 phase lag, assuming the larger M2 amplitude of 50.29 cm and varying the phase lag obtained from tidal analysis of the 1984 data in increments of 1°, and computing the average difference in the times of high and low waters between those in the Ross data set and the predictions (Table 3b). The HW values suggest that the historical phase lag was 1° larger in 1842 than in 1984, while the LW values suggest a 2° larger historical phase lag. Taking into account the uncertainties in timings that are
inherent in all tide gauge records, we can conclude that these findings are consistent with zero phase lag difference.

### 3.2.3 Explanation of a Larger M2 Amplitude using the HL Tabulations

An explanation of why the M2 amplitude calculated from the HL tabulations in Sects. 3.2.1 and 3.2.2 is larger than that found from the nineteenth-century harmonic analysis by Edward Roberts (Sect. 3.1) is as follows.
Whewell and his assistants extracted tables of HL values for many stations around the world as part of their research on the tides. In the present case, the only feasible way that the tables for Port Louis could have been made is by inspection of the several quarter-hour values obtained by Ross around each expected astronomical high and low water. Whewell could only have picked out the highest (lowest) value in each group and entered that into the table as being the actual observed high (low) water value. He would not have had time, or the
computational ability, to have put a smooth spline through the noise in the several quarter-hour measurements in each group and so arrive at an interpolated height estimate.

            In practice, one could imagine that high-frequency, non-tidal variability could have resulted in one of the quarter-hourly values either side of a tidal HL being higher (lower) than that at the time of astronomical tidal HL, and so being tabulated as the actual, observed HL value. This would be the case if, for example, there was
considerable variability on timescales of an hour or two, and it would result in estimates of tidal MHW being biased slightly too high and tidal MLW being biased slightly too low. This suggestion is illustrated schematically in Figure 3.

            In support of this theory, one can look at the power spectrum of the 1984 record which shows a broad peak of energy centred on 0.5 cph (Figure 4), consistent with the seiche-type signals to be seen in all of the Port
Louis records, with amplitudes of several cm and periods of about 2 hours. The root-mean-square of variability within this band (0.3-0.65 cph) is 3.5 cm The suggestion that this variability is due to a seiche in Berkeley





Sound is supported by knowledge of its dimensions (approximately 31.3 km long and 50 m deep at the mouth, see UKHO, 1989) and with the use of the 'rectangular/rectangular' formula of Wilson (1972, page 40) for a quarter-wavelength resonance with maximum amplitude at the head of the inlet:


$$T = 2.618 \, (2 \, L) \, / \, \sqrt{(g \, h)}$$

where L is the length of the inlet, h the depth at its mouth, g acceleration due to gravity and T is the period of the resonance. The latter is thereby calculated to be 2.03 hours, or a frequency of 0.492 cph.

195          As a test, each of the 1984 half-hourly values nearest to high (or low) astronomical tide were inspected to see whether there was another half-hourly level within one hour which was larger (smaller) than this one. If this one had indeed been the largest (smallest) of the group of five considered, then one could have imagined Whewell selecting that one as the official HL value straightforwardly. But if one if the other four in the group was larger (smaller) because of the high-frequency variability, then that one would have been selected for 200   tabulation instead. In fact, the test showed that there would have been biases of 2.9 cm (upwards) and 2.6 cm (downwards) in the calculation of MHW and MLW respectively, or 5.5 cm in MTR and so 2.7 cm in M2 amplitude. Now, Whewell would have had quarter-hourly values available, not half-hourly, so one would expect any biases from his tabulations to be smaller than this. But the general idea is the same, and it is feasible to expect that the use of his tabulations would have resulted in positive biases in M2 amplitude of ˜1 cm.

**3.3 Comparison to Modern Values of M2**

Table 2 shows the findings for M2 from tidal analyses of the modern data sets; listings of all constituents from the analyses can be found in the Supplementary Material. The analyses took the form of conventional harmonic analysis with a number of constituents appropriate for the record lengths, longer records enabling the amplitudes and phase lags of more constituents to be determined independently (Bell et al., 1999; Pugh and Woodworth, 210   2014). For the shortest records (numbers 5 and 6 in particular), some semidiurnal constituents cannot be determined independently of M2 and so their amplitudes have to be specified in terms of 'related constituents' in the harmonic fit. For example, N2 has an amplitude of 0.1915 of that of M2 in the equilibrium tide, and that is indeed found at many locations around the world. However, analysis of records 1, 2, 3 and 4 and the historical analysis of Baird and Darwin (1885) indicated that at Port Louis the ratio is closer to 0.25 (Table 2). Therefore, 215   the relationships for N2, NU2 and L2 relative to M2, obtained from the analysis of the longer record 2, were employed in the tidal analyses of records 5 and 6.

         The average of the resulting six amplitudes of M2 in Table 2 is 47.5 cm, with a scatter between them of 0.5 cm. Such a small scatter is to be expected, as M2 is known to fluctuate from year to year depending on varying meteorological and oceanographic forcings (cf. Figure 2a for Port Stanley and see Pugh and 220   Woodworth, 2014), and given that measurements were obtained using different tide gauges and at slightly different locations. Nodal variation is accommodated automatically in the tidal analyses, so the individual amplitudes and their average do not have to be adjusted further for this variation. Seasonal adjustments might increase the average by several mm, as records 1, 5 and 6 in particular are short and occur at the start of the year; that would raise the average M2 amplitude to approximately 48 cm, which is almost identical to that for 225   1842 in the ATT tables calculated from Ross's data (Sect. 3.1). The average Greenwich phase lag is 276.4° with





two individual values within 1° of the average and all within 2°. This scatter is actually smaller than that which might be expected from the 0.5 cm scatter in the amplitudes and so can be considered as being rather small indeed.

**3.4 Comparison to Modern Values of S2, K1 and O1**

Average values for the amplitudes and Greenwich phase lags of the other main semidiurnal constituent (S2) and the two main diurnal constituents (K1 and O1) can also be obtained from the six modern records:

| Constituent | H(cm) | G(°) |
|---|---|---|
| M2 | 47.5 | 276.4 |
| S2 | 16.9 | 311.9 |
| K1 | 12.7 | 101.8 |
| O1 | 15.1 | 57.8 |


which indicates differences of only 1-2 cm in amplitude and several degrees in phase lag for S2, K1 and O1 from those reported above for 1842 in the ATT tables (Sect. 3.1). Such small differences in these smaller constituents, in phase lag in particular, cannot be regarded as significant given the year-to-year variability in the ocean and in measurement techniques.

**4. Conclusions**

The modern values for M2 (amplitude 48 cm and 276.4° phase lag) can be compared to those obtained from harmonic analysis of the 1842 data as reported by Baird and Darwin (1885) and included in the ATT tables, which indicated an M2 amplitude of 47 cm and Greenwich phase lag of 273°. In addition, analyses of the MTR tabulations described in Sects. 3.2.1 and 3.2.2 suggest an M2 amplitude of 49 or 50 cm, which may be over-estimated by ˜1 cm for the reasons given in Sect. 3.2.3. Meanwhile, the phase lag analysis of Sect. 3.2.2

suggests that the M2 phase lag in 1842 was only 1 or 2° larger than the 277.1° in 1984.

Therefore, the main conclusion of this study is that there is no evidence that the ocean tide at Port Louis has changed significantly since 1842; almost no change in amplitude and only several degrees in phase lag at most. This contrasts with findings elsewhere, and especially in the northern hemisphere, where significant changes have been observed (Woodworth, 2010; Haigh et al., 2020). However, this could, of course, be simply

part of a complicated worldwide pattern of tidal changes. It will be particularly interesting to combine the findings of the present study with those from elsewhere in the region; for example, tides measured by the French Expedition to Orange Bay near Cape Horn in the early 1880s as part of the First International Polar Year are presently under investigation (Laurent Testut, private communication). On a more technical level, the present study has also been useful in providing an example of obtaining accurate tidal information from tabulations of

HL waters and from short tide gauge records.



**Supplementary Material.** The archive 'supplement.zip' includes one file containing the 1842 Ross HL values, six containing the modern records of Table 1, one containing full listings of the tidal analyses of the 6 records and a 'readme' information file.


**Competing interests.** The author declares that he has no conflict of interest.

**Author contribution.** PLW performed the analyses described above and wrote the paper.

**Acknowledgements.** I would like to record the contributions of my colleague David Pugh who died in 2022.
David instigated or undertook most of the measurements in Table 2. I had great pleasure in going on fieldwork to the Falklands with him in 2009 during which records 5 and 6 in Table 1 were obtained. I am grateful to many colleagues at the NOC and Ross Chalenor and colleagues at the Falkland Islands Public Works Department for all their help. Christopher Jones of the UKHO is thanked for information and advice about the 1981-2 tide gauge data and many other matters. Also I am grateful to Paul Hughes (formerly Liverpool John Moores University)
who discovered the Ross high and low water records in the Wren Library. Laurent Testut is thanked for comments on an earlier version of this paper. This work was funded partly by the U.K. Natural Environment Research Council. Some figures were generated using the Generic Mapping Tools (Wessel and Smith, 1998).



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



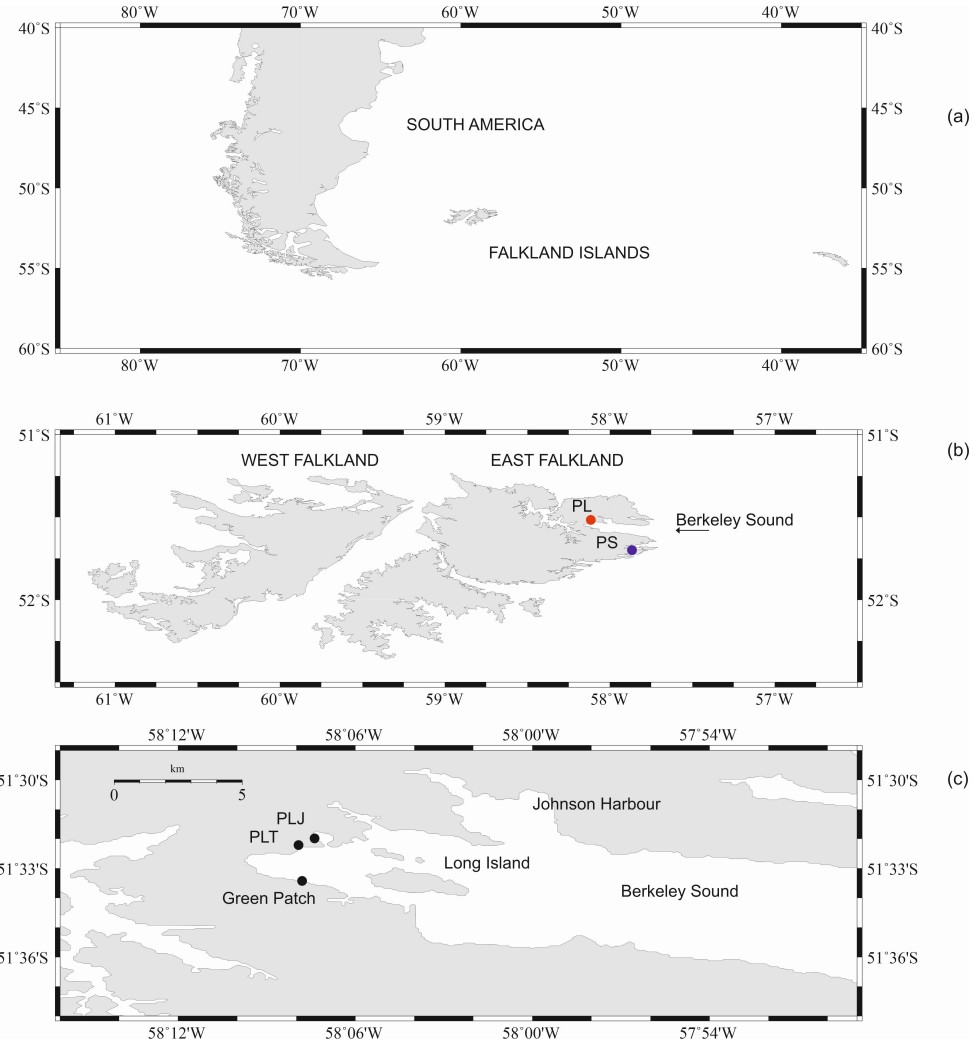


Figure 1. (a) Relationship of the Falkland Islands to the coast of South America in the SW Atlantic Ocean, (b) Map of West and East Falkland showing the positions of Berkeley Sound and Port Louis (PL, red dot) and Port Stanley (PS, blue dot), (c) Locations of Port Louis Jetty (PLJ), which is in a small inlet, Port Louis Tablet (PLT) and Green Patch at the head of Berkeley Sound.




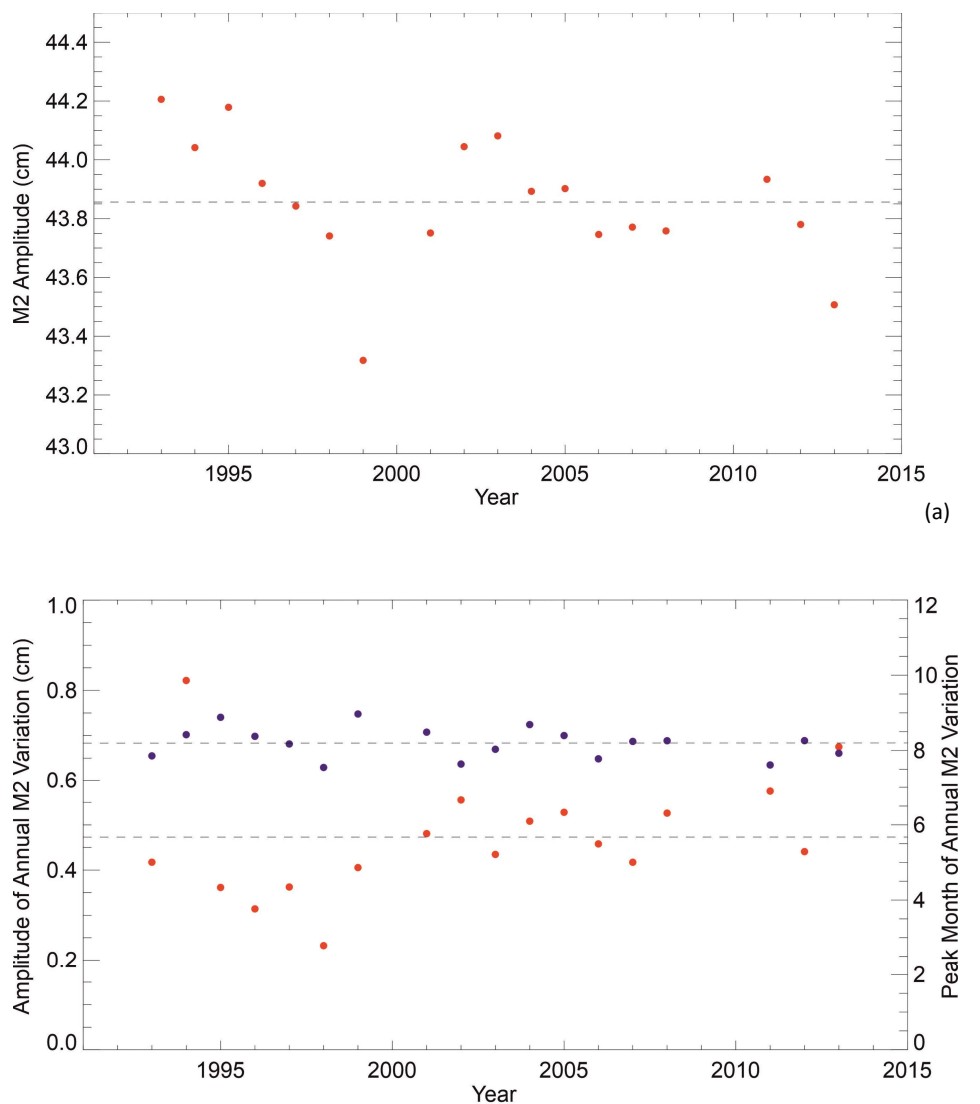

(a)

,b)

Figure 2. (a) Amplitude of the M2 component of the ocean tide at Port Stanley, 25 km southeast of Port Louis, computed from 18 separate annual tidal analyses between 1993 and 2015, each adjusted for nodal variation as in the equilibrium tide. Values fluctuate from year to year about the average (dashed line) but no residual nodal signal is evident. (b) Amplitude (red dots, left hand scale) and peak month (blue dots, right hand scale) from the 18 separate annual tidal analyses. The average of each quantity is shown by the dashed lines.





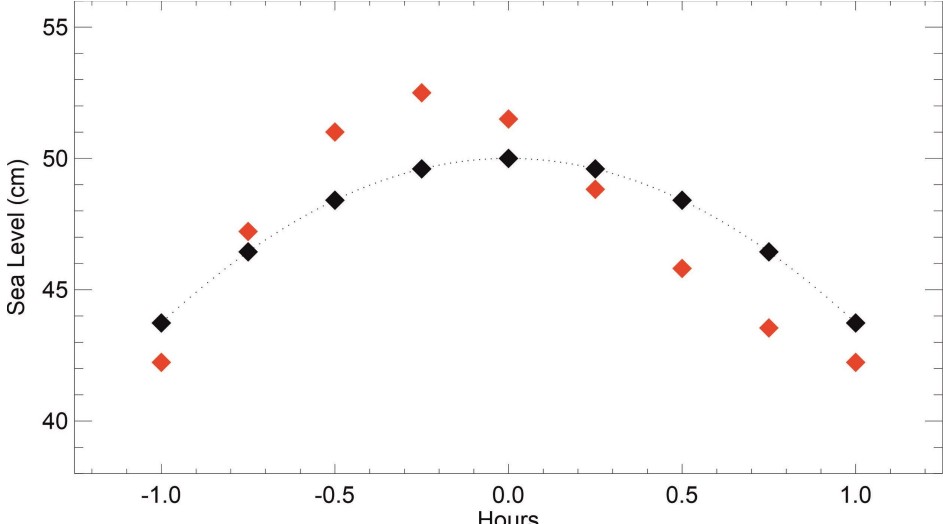

Figure 3. A schematic example of how high water tabulated by Whewell might have been biased high because of high-frequency variability. The black curve shows a tide with an amplitude of 50 cm around
the time of high tide, peaking at time zero. A curve similar to this occurs if the only processes involved are the astronomical tides, low-frequency storm surges and changes in mean sea level. In these cases, 15-minute measurements will sample the curve as shown by the black diamonds, and the highest of these will define the recorded high water. But if there is also sufficient high-frequency variability present on timescales shorter than the flattish upper part of the curve, simulated here by an oscillation
with a period of 2 hours and amplitude of 3 cm peaking a short time before time zero, there would have been some occasion in the flattish part when there was a red diamond (the total level) that was higher than the highest black one. In this case, Whewell would have tabulated the highest red value as the high water.



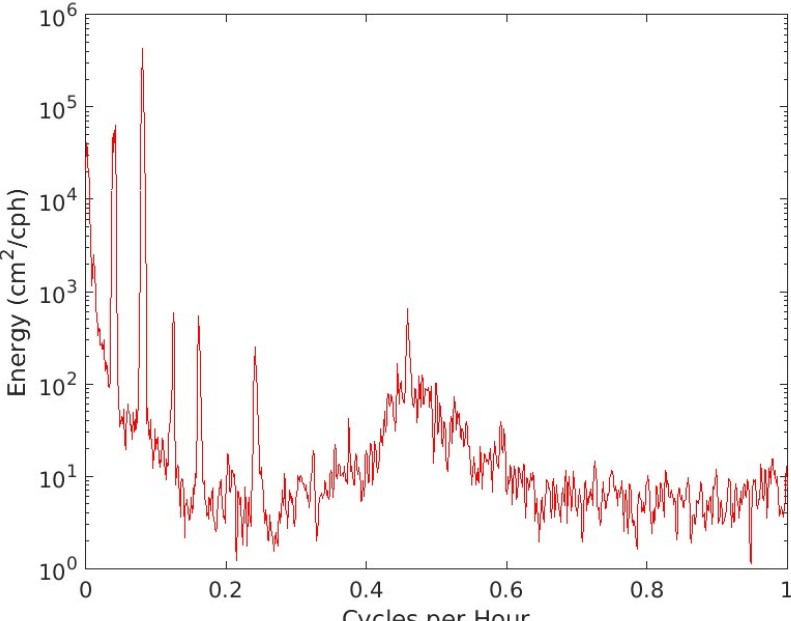

Figure 4. Power spectrum of the half-hourly 1984 sea level record (record 2 in Table 2). The spikes on the left
are due to the main diurnal and semidiurnal tides and their harmonics, while the broad enhancement centred
around 0.5 cpd is due to the seiche activity in Berkeley Sound.




Table 1. Recent observations of the ocean tide from tide gauges at or near to Port Louis in Berkeley Sound, East Falkland showing durations of the measurements, locations (Figure 1), gauge types and recording frequencies.

| Record | Duration (years/days) | Location | Gauge Type | Rec. Freq. (min) |
|---|---|---|---|---|
| 1. | 1981/339 – 1982/87 | Green Patch | Float gauge | 30 |
| 2. | 1984/143-252 | Port Louis Jetty | Bubbler pressure gauge | 30 |
| 3. | 2003/110-265 | Port Louis Jetty | Absolute pressure gauge | 6 |
| 4. | 2003/110-265 | Green Patch | Absolute pressure gauge | 6 |
| 5. | 2009/55-70 | Port Louis Tablet | Absolute pressure gauge | 6 |
| 6. | 2009/55-70 | Port Louis Jetty | Differential pressure gauge | 1 |

Table 2. Amplitude (H) and Greenwich phase lag (G) of the M2 component of the ocean tide determined from the tide gauge records in Table 1. The number of major and related harmonic constituents used in each analysis is shown together with analysis comments.

| Record | Amplitude H (cm) | Phase Lag G (°) | No. Major Con. | No. Rel. Con. | Comments |
|---|---|---|---|---|---|
| 1. | 47.5 | 277.9 | 27 | 8 | N2/M2 = 0.261 |
| 2. | 48.0 | 277.7 | 55 | 2 | N2/M2 = 0.245 |
| 3. | 47.7 | 277.3 | 55 | 2 | N2/M2 = 0.253 |
| 4. | 47.6 | 275.5 | 55 | 2 | N2/M2 = 0.254 |
| 5. | 47.1 | 274.4 | 16 | 15 | N2, NU2 and L2 related to M2 as determined in the 1984 analysis |
| 6. | 46.9 | 275.3 | 16 | 15 | ditto |



Table 3a. The difference between MTR measured in 1842 minus that predicted using 1984 data, incrementing the M2 amplitude used in the predictions (48.04 cm) by small amounts.

| Amplitude Increment (mm) | MTR Difference (cm) |
|---|---|
| 0 | 4.14 |
| 10 | 2.31 |
| 15 | 1.39 |
| 20 | 0.47 |
| 21.5 | 0.20 |
| 22 | 0.10 |
| 22.5 | 0.01 |
| 23 | -0.08 |
| 23.5 | -0.17 |
| 24 | -0.27 |
| 25 | -0.45 |

Table 3b. Average difference (min) between the times of high and low waters in the Ross data set and those in predictions for 1842 based on the 1984 data set, considering increments (°) on the M2 phase lag obtained in 1984 (277.1°).

| Phase Lag Increment (°) | Time Diff. (min) HW | Time Diff. (min) LW |
|---|---|---|
| -2 | 5.35 | 8.20 |
| 0 | 1.49 | 4.42 |
| 1 | -0.53 | 2.54 |
| 2 | -2.41 | 0.48 |
| 3 | -4.42 | -1.51 |
| 4 | -6.48 | -3.40 |