# Peer review of "Long-term changes in the ocean tide at Port Louis, Falkland Islands"

_EGUsphere, 2024_

## Author Response (AR1)

14 May 2024

Dear Sir,

Here are my point-by-point responses to the review comments on this paper. Each comment is followed by my response in capitals. I am grateful to both reviewers for the time spent on this.

There then follows a mention that the tables have been renumbered as requested by the editorial office. The tables were anyway out-of-order in the original submission so had to be renumbered.

In addition, the text has had a few small edits which will be clear from inspection of the track changes in the Word comparison file.

Many thanks once again for the help with this.

Philip Woodworth

RC1: Christopher Jones

Line 198, minor typo:

"But if one if the other four in the group"

should read

"But if one of the other four in the group"

FIXED.

RC2: Anonymous Reviewer

In this paper the author sets out to compare the tidal information in Port Louis (Falklands) reported on by Ross in the 1800 with contemporary data to see if there has been any change. After reading the paper through to results my initial thought was "so what have we learned?". The discussion, however, swings things around and there is a good context for the work there which I suggest is highlighted further in the introduction. Apart from showing that the tide in the area hasn't changed much, which is of some interest, the methodology presented makes this worth publishing. Consequently, I recommend publication after very minor revisions.

Comments:

As mentioned above, I would like a better motivation in the introduction, including all points made in the discussion. The methods is what is most valuable here, in my mind.

Minor point: Do we really have enough accuracy in the data, especially the 19th century data, to talk about changes on mm scale?  A comment on accuracy of the measurements would be good.

I AGREE THAT THE METHODS USED ARE IMPORTANT, AS THE REVIEWER SAYS, AND I HAVE TAKEN HIS ADVICE TO HIGHLIGHT THEM FURTHER IN THE INTRODUCTION IN A REVISED VERSION. HOWEVER,

THE RESULT (OR NULL RESULT ONE MIGHT SAY) ON CHANGES IN THE TIDE IS THE MOST IMPORTANT THING, GIVEN THAT CHANGES IN TIDES HAVE BEEN REPORTED ELSEWHERE.

AS FOR THE ACCURACY OF THE ROSS MEASUREMENTS, IT IS INTERESTING THAT BY MAKING SEA LEVEL MEASUREMENTS USING EYEBALL READINGS OF A TIDE POLE (OR STILLING WELL), AS ROSS WILL HAVE DONE, ONE CAN ESTIMATE TIDAL PARAMETERS ACCURATELY (SAY TO THE CENTIMETRIC LEVEL). THAT IS BECAUSE HARMONICS SEPARATE OUT FROM A NOISY TIME SERIES. OR AT LEAST, THAT WOULD HAVE BEEN THE SITUATION IF ROSS'S ORIGINAL HALF-HOURLY VALUES HAD SURVIVED. THE FACT THAT THEY HAVE NOT, BUT THAT WE HAVE ONLY THE HIGH AND LOW WATER VALUES AVAILABLE, ADDS AN EXTRA COMPLICATION WHICH IS DESCRIBED IN THE PAPER. I HAVE ADDED A PARAGRAPH TO EXPLAIN THIS MORE. BUT WE ARE NOT CLAIMING MM CONSISTENCY, AS THE AUTHOR SUGGESTS, THE OLD AND NEW MEASUREMENTS COMPARE AT APPROXIMATELY THE CM LEVEL AS SUMMARISED IN THE CONCLUSIONS.

Editorial Request

AS REQUESTED IN EMAIL OF 21 FEBRUARY FROM THE EDITORIAL OFFICE, TABLE 3(A,B) WAS SPLIT INTO TWO. THESE ARE NOW CALLED TABLES 2 AND 3 AND THE OLD TABLE 2 IS NOW TABLE 4.